# Genetic polymorphisms of superoxide dismutase 1 are associated with the serum lipid profiles of Han Chinese adults in a sexually dimorphic manner

**Ping Xu[1]☯, Yumei Zhu[2]☯, Xiongshun Liang[2], Chunmei Gong[2], Yuanfei Xu[2], Changhua Huang[3], Xiao-Li Liu[2], Ji-Chang Zhou**[1,2,4,5]*

1 School of Public Health (Shenzhen), Sun Yat-sen University, Shenzhen, Guangdong, China, 2 Shenzhen Center for Chronic Disease Control, Shenzhen, Guangdong, China, 3 Shenzhen Qilinshan Sanatorium, Shenzhen, Guangdong, China, 4 Guangdong Provincial Key Laboratory of Food, Nutrition and Health, Guangzhou, Guangdong, China, 5 Guangdong Province Engineering Laboratory for Nutrition Translation, Guangzhou, Guangdong, China

☯ These authors contributed equally to this work.
* zhoujch8@mail.sysu.edu.cn

**Data Availability Statement:** The epidemiological data sheet is available on Mendeley (DOI: 10.17632/7cg9ngb8zw.2).

## Abstract

Inspired by the mechanistic correlations between superoxide dismutase 1 (SOD1) and lipid metabolism, the associations of *SOD1* single nucleotide polymorphisms (SNPs) with circulating lipid levels were explored. In 2621 Chinese Han adults, randomly recruited from a health examination center without organic diseases, cancers, and pregnancy, three tag SNPs, rs4998557, rs1041740, and rs17880487 selected by Haploview software were genotyped with a probe-based real-time quantitative PCR method. In both genders, most parameters of the dyslipidemia adults were inferior ($P < 0.001$) to those of the non-dyslipidemia adults, and genotype frequencies of rs4998557 and rs17880487 were significantly different ($P < 0.05$) between the normal and abnormal subgroups of total cholesterol (TC) or high-density lipoprotein cholesterol (HDLC). Adjusted for confounding factors, logistic regression analyses revealed that in males rs4998557A, rs1041740T, and rs17880487T reduced the risk of high TC and/or LDLC ($P < 0.05$), and rs4998557A and rs17880487T increased the risk of low HDLC ($P < 0.05$); but in females, none of the SNPs had associations with any of the lipid parameters ($P > 0.05$). Conclusively, characterized by a sexual dimorphism, the *SOD1* polymorphisms were associated with the lipid disorders in the adult males but not females of the Chinese Han population.

## Introduction

Dyslipidemia is one of the most prevalent health problems in the modern era, and multiple factors are thought to be the etiology [1, 2]. One of them is the dysfunction of antioxidant system, which leads to an increase in the production and a decrease in the inactivation of reactive oxygen species (ROS) [3]. Oxidative stress is produced once the production of ROS

**Funding:** The study was partly supported by the National Natural Science Foundation of China (grant number 81973038 to JCZ) and Healthcare Research Projects of Shenzhen, China (grant number SZGW2017010 to JCZ and SZGW2017015 to CG). The funders had no role in study design, data collection and analysis, decision to publish, or preparation of the manuscript. Website of the National Natural Science Foundation of China: http://www.nsfc.gov.cn/. Website of the Shenzhen Municipal Health Commission: http://wjw.sz.gov.cn/. These were all the sources of funding supporting this study, and there was no additional external funding received for it.

**Competing interests:** The authors have declared that no competing interests exist.

overwhelms the antioxidant capacity. Superoxide dismutases (SODs) are a ubiquitous class of antioxidant metalloproteinases, consisting of a total of three genetically distinct isoforms in human [4]. Superoxide dismutase 1 (SOD1, EC: 1.15.1.1), a copper-and zinc-containing SOD located at the cytoplasm, nucleus, mitochondrial intermembrane space as well as serum lipoproteins [4–7], accounts for 50% to 90% of the total SOD activity in a eukaryotic cell or mammalian tissues [4, 6, 8] and plays a key role in the maintenance of a physiological ROS level by catalyzing superoxide anion ($O_2^{\bullet-}$) to hydroperoxide and oxygen [9]. Supraphysiological levels of ROS are extremely detrimental to DNA, lipids, proteins, and normal cellular metabolism [10, 11], and have a strong potential to disturb the lipid metabolism [12]. The *Sod1* knockout mice were characterized by lipid accumulation in liver and abnormal circulating lipid profiles [13, 14], and the inhibition of SOD1 function in nasopharyngeal carcinoma connived the accumulation of lipid droplets [15]. SOD1 also affected cholesterol metabolism in human hepatocarcinoma cells [16] and its presence in human serum lipoproteins suggested its crucial role in the lipid transport [7].

The single nucleotide polymorphism (SNP) is the most common type of DNA variations in > 1% of a population [17], usually expressed as its minor allele frequency (MAF) > 0.01. The SNPs in a gene may change the gene activities, alter the amino acid residues, moderate protein functions, and/or exert some other effects on the molecular level to ultimately affect the phenotypes [18]. Several SNPs of *SOD1* have been reported to correlate with metabolic disorders such as obesity [19], diabetes and its complications [20–24], cardiovascular disease [25, 26], etc., but their associations with lipid profiles and dyslipidemia were absent [27]. On the other hand, accumulating evidences revealed that males and females usually exhibit sex-specific differences in susceptibility, prevalence, morbidity, symptoms, treatment, or prognosis for many diseases, and females maybe more resistant to oxidative damage [28]. In terms of oxidative stress regulation pathways, it has been found that the potential sexual dimorphism may have diverse effects on the cardiovascular diseases in the two genders [29]. As a gene encoding an important antioxidant enzyme, *SOD1* may have its SNPs gender-differentially correlated with metabolic diseases [30].

Though the previous studies revealed that the knockout/inhibition of *SOD1* induced disorders of lipid metabolism [3, 13–16] and the SNPs of *SOD1* may correlate with some metabolic disorders [19–21, 25, 26, 31, 32], no epidemiologic studies were yet conducted to investigate the association of *SOD1* SNPs and dyslipidemia, not to mention the effect of gender differences on the association. Therefore, the objective of this study was trying to understand the situation.

## Materials and methods

### Participants and sample collection

The study was approved by the Ethic Committee of Shenzhen Center for Chronic Disease Control, and registered in January 2018 as NCT03406234 in the ClinicalTrials.gov online system as reported in our previous study [33]. Adult volunteers were recruited from a health examination center in Shenzhen city of Guangdong province, China. They were informed about the study and their privacy right protection with written consents to be the participant candidates and accepted a questionnaire survey on their basic health information. The anthropometric measurements for height and body weight were performed on every volunteer. The inclusion criteria were: 1) ≥ 18 years old Han Chinese; 2) having been living in Shenzhen for > 2 years; 3) free of any type of cancers and other organic diseases in the past 6 months according to their questionnaires and medical records; 4) not taking long-term effect medicines to control lipid profiles; and 5) not in pregnancy for women.

The sample size (N) was estimated with the formula $N \geq deff \times Z^2 \times p(1-p)/E^2$, where deff was 2.0 for the design effect, Z was 1.96 for the two-sided 95% confidence intervals (CIs), p was 8.15% for the prevalence of high total cholesterol (TC) in Chinese adult males [34], E was 20% of the prevalence for the relative precession, and the calculated N was $\geq$ 2165. Furthermore, the sample size was confirmed by the QUANTO software to be sufficient to detect the genetic difference. In the independent individuals and gene only model, assuming the minimum MAF = 0.01 and odds ratio (OR) = 0.1, 781 was the minimum number of sample size for each gender. Based on the estimated total sample size and a gender ratio close to 1, at least 1083 (2165/2) for each gender were required.

The people visiting the health examination center were fasted overnight and did not take medicines to control the previously diagnosed dyslipidemia for more than 12 hours. From their ulnar veins, blood samples were drawn into a vacuum tube with EDTA anticoagulant and another vacuum tube without any anticoagulants, respectively. After the immediate assays for routine blood parameters, the whole blood samples were centrifuge at $3,000 \times g$, 4˚C for 10 minutes to get supernatant and precipitate (mainly blood cells). The separated sections of each blood sample were transferred to aliquot tubes for timely analysis or stored at -80˚C for later experiments. Lipid profiles, fasting plasma glucose (FPG), and some other biomarkers for liver and renal functions were assayed less than two hours after the blood sample collection. The body weight and height were measured to calculate the body mass index (BMI) as body weight (kg)/ height (m)$^2$. Blood pressures were measured with certified mercury blood-pressure meters. Later, referring to the diagnosis criteria recommended by the Chinese guideline for dyslipidemia management [35], a subject was grouped into the dyslipidemia group if he/she had triglyceride (TG) $\geq$ 2.3 mmol/L, TC $\geq$ 6.2 mmol/L, low-density lipoprotein cholesterol (LDLC) $\geq$ 4.1 mmol/L, high-density lipoprotein cholesterol (HDLC) $<$ 1.0 mmol/L, or was a previously diagnosed dyslipidemia patient; otherwise, he/she was grouped into the non-dyslipidemia group. In the genotype frequency analyses for either subgroup of TG, TC, or LDLC, in order to avoid the small subject numbers for the genotypes of low MAF, the participants were grouped as the abnormal subgroup by the available marginally elevated cut-off values for TG $\geq$ 1.7 mmol/L, TC $\geq$ 5.2 mmol/L, or LDLC $\geq$ 3.4 mmol/L [35], respectively, or otherwise as the corresponding normal subgroup for each of the lipid parameters. Subgroups of abnormal and normal HDLC were defined by the only available cut-off value of 1.0 mmol/L [35].

## DNA preparation

The genomic DNA was extracted from each of the blood cell samples according to the user manual of the commercial kit (QIAGEN Cat#: 51106). The DNA concentration was assayed with a spectrometer (NanoVue Plus, GE), and diluted with double-distilled water to the final concentration of 100 ng/μl for later SNPs analyses.

## Selection and genotyping of tag SNPs

The files for SNP data of *SOD1* were downloaded from The International Genome Sample Resource (IGSR) (http://www.internationalgenome.org/). Three SNPs of rs4998557, rs1041740, and rs17880487 capturing the total alleles at $r^2 \geq 0.8$ were selected as tag SNPs by the Haploview 4.2 software (S1 Fig).

For each of the SNPs, specific primers and molecular beacon probes were designed with Primer Premier 5 software, and synthesized by the Invitrogen Ltd. (Shanghai, China). S1 Table summarized the information of the oligos and the amplicons. The SNPs were analyzed on the LightCycler 480 II real-time quantitative PCR (qPCR) machine (Roche, Singapore). A hot-start Taq enzyme kit (Cat#: DR007B, TaKaRa, China) was used to perform the qPCR reaction.

The asymmetric PCRs were performed to genotype the SNPs. In brief, in each of the 25 μl well of the 96-well plate, the synthesis of the probe-targeted strand was initiated by the primer having 10 times of concentration to that of the other primer. The qPCR program was: 1) 94ºC for 3 min; 2) a touchdown step of 10 cycles of 94ºC for 15 sec, 65ºC (decreasing at 1ºC for each cycle) for 15 sec, 72ºC for 20 sec; 3) 50 cycles of 94ºC for 15 sec, 55ºC for 15 sec of signal collection, and 72ºC for 20 sec; 4) a melting curve step of 94ºC for 1 min, 40ºC for 3 min, and a temperature increase from 40ºC to 80ºC with a collection of 5 points of signals per degree. The genotypic polymorphisms of major homozygote, minor homozygote, and heterozygote judged by the curve pattern with peak(s) at specific melting temperature(s) were verified with Sanger sequencing analyses by the Invitrogen Ltd. (Shanghai, China). For each of the three SNPs' amplicons, the sequencing primer, as indicated in S1 Table, was one of the primers for the above qPCR amplification. Consequently, all the genotyping work was performed with our established molecular beacon probe-based qPCR method.

### Data analysis

Data of clinic profiles, such as anthropometric indices, fasting glucose and lipid levels, biomarkers for liver and renal functions, routine blood parameters, etc., of the non-dyslipidemia and dyslipidemia adults were presented as means ± SD and analyzed with $t$ test between the two groups in either gender. The genotypes of all the three SNPs were tested with Hardy-Weinberg equilibrium (HWE) analyses for sampling representation. The lipid parameters grouped by the three tag SNPs of *SOD1* were expressed as medians and their interquartile ranges, and compared with rank sum test (Wilcoxon rank test and Kruskal-Wallis rank test). For genotypic comparisons, differences in allele and genotype frequencies were evaluated using the Chi-square ($\chi^2$) test. The additive, dominant, recessive, homozygous, or allelic models for each of the SNPs entered the logistic regression analyses for ORs and 95% CIs with adjustment for age, BMI, education (elementary school, junior high school, senior high school, undergraduate, or postgraduate), FPG, and smoking status (currently daily, currently occasional, former, or never). A multiple comparison test was performed when there was a significant difference among at least three groups. The $P$-value less than 0.05 was considered to be statistically significant.

## Results

There were 1110 adult males and 1511 adult females (2621 in total) included for the study, and the clinic profiles of the participants were summarized in Table 1. It was indicated that several metabolic or metabolism-related parameters were statistically different between non-dyslipidemia and dyslipidemia adults ($P < 0.05$), such as age, BMI, systolic blood pressure, diastolic blood pressure, FPG, serum levels of TG, TC, LDLC and HDLC, alanine aminotransferase, aspartate aminotransferase, and so forth. The composition of dyslipidemia in male and female subjects was showed in S2 Table.

Human *SOD1* is a gene spanning about 9310 base pairs, and has five SNPs for Chinese marked by the Haploview software (S1 Fig). Rs4998557 and rs2070424 have a linkage disequilibrium (LD) degree of 0.97, and rs1041740 and rs4817420 have a LD of 1.00. Whereas rs17880487 had low LD with any of the other SNPs. Thus, rs4998557, rs1041740, and rs17880487 were selected as tag SNPs to represent the polymorphisms of *SOD1*. The melting curves representing the major homozygotes, minor homozygotes, and heterozygotes of the three SNPs and the sequencing verification were shown in S2–S4 Figs. None of the frequencies of major homozygote, heterozygote, minor homozygote, major allele, or minor allele of all the

**Table 1. Clinic profiles of the non-dyslipidemia (ND) and dyslipidemia (DL) adults, mean ± SD.**

| | Male, *n* = 1110 | | *t*-value | *P*-value | Female, *n* = 1511 | | *t*-value | *P*-value |
|---|---|---|---|---|---|---|---|---|
| | ND, 66.7% | DL, 33.3% | | | ND, 79.9% | DL, 20.1% | | |
| Age, y | 36.0 ± 10.2 | 40.2 ± 10.0 | 6.527 | < **0.001** | 37.9 ± 11.2 | 46.6 ± 13.3 | 10.539 | < **0.001** |
| BMI, kg/m$^2$ | 23.9 ± 2.8 | 25.5 ± 2.9 | 8.707 | < **0.001** | 21.7 ± 2.6 | 23.0 ± 2.9 | 7.527 | < **0.001** |
| SBP, mmHg | 122.6 ± 14.3 | 126.5 ± 13.5 | 4.451 | < **0.001** | 113.1 ± 14.6 | 122.1 ± 18.7 | 7.852 | < **0.001** |
| DBP, mmHg | 74.1 ± 10.3 | 77.7 ± 9.9 | 5.577 | < **0.001** | 66.8 ± 9.5 | 71.1 ± 11.4 | 5.993 | < **0.001** |
| FPG, mmol/L | 5.4 ± 0.7 | 5.8 ± 1.4 | 5.195 | < **0.001** | 5.3 ± 0.6 | 5.8 ± 1.3 | 6.755 | < **0.001** |
| TG, mmol/L | 1.2 ± 0.5 | 3.3 ± 2.6 | 15.220 | < **0.001** | 0.9 ± 0.4 | 2.1 ± 2.0 | 10.724 | < **0.001** |
| TC, mmol/L | 4.9 ± 0.7 | 5.5 ± 1.1 | 10.238 | < **0.001** | 4.8 ± 0.7 | 6.0 ± 1.4 | 14.814 | < **0.001** |
| LDLC, mmol/L | 2.8 ± 0.4 | 3.2 ± 0.6 | 10.564 | < **0.001** | 2.7 ± 0.4 | 3.5 ± 0.8 | 15.327 | < **0.001** |
| HDLC, mmol/L | 1.3 ± 0.2 | 1.4 ± 0.2 | 5.803 | < **0.001** | 1.4 ± 0.2 | 1.5 ± 0.3 | 6.185 | < **0.001** |
| ALT, IU/L | 27.2 ± 28.5 | 33.4 ± 19.2 | 4.311 | < **0.001** | 18.1 ± 9.7 | 21.5 ± 11.0 | 4.995 | < **0.001** |
| AST, IU/L | 23.6 ± 15.3 | 25.7 ± 9.4 | 2.792 | < **0.001** | 19.4 ± 6.1 | 21.4 ± 6.6 | 4.743 | < **0.001** |
| DB, μmol/L | 5.3 ± 1.2 | 5.2 ± 1.2 | -1.457 | 0.145 | 4.9 ± 1.2 | 4.8 ± 1.1 | -0.824 | 0.410 |
| TB, μmol/L | 17.9 ± 5.9 | 17.3 ± 5.8 | -1.627 | 0.104 | 15.0 ± 5.2 | 13.9 ± 4.1 | -3.895 | **0.001** |
| Cr, μmol/L | 90.4 ± 13.3 | 91.5 ± 15.5 | 1.225 | 0.221 | 66.0 ± 9.7 | 67.9 ± 11.6 | 2.610 | **0.004** |
| UA, μmol/L | 379.1 ± 72.8 | 412.1 ± 77.1 | 6.853 | < **0.001** | 271.0 ± 55.1 | 306.1 ± 68.2 | 8.321 | < **0.001** |
| UN, mmol/L | 4.5 ± 1.2 | 4.6 ± 1.2 | 1.335 | 0.182 | 4.0 ± 1.1 | 4.3 ± 1.2 | 4.309 | < **0.001** |
| TP, g/L | 70.4 ± 3.8 | 70.5 ± 3.8 | 0.230 | 0.819 | 69.8 ± 3.8 | 70.1 ± 3.7 | 1.184 | 0.237 |
| ALB, g/L | 44.5 ± 2.7 | 44.1 ± 2.7 | -2.256 | **0.026** | 42.9 ± 2.7 | 42.6 ± 2.6 | -1.458 | 0.145 |
| Hb, g/L | 147.5 ± 9.8 | 149.4 ± 10.0 | 3.042 | **0.002** | 126.3 ± 10.3 | 129.8 ± 9.6 | 5.490 | < **0.001** |
| Platelet, 10$^9$/L | 215.8 ± 44.1 | 224.0 ± 43.7 | 2.946 | **0.003** | 239.5 ± 51.8 | 245.3 ± 53.3 | 1.737 | 0.083 |
| RBC, 10$^{12}$/L | 5.1 ± 0.4 | 5.1 ± 0.4 | 1.744 | 0.081 | 4.4 ± 0.4 | 4.5 ± 0.4 | 2.376 | **0.018** |
| WBC, 10$^9$/L | 6.8 ± 1.6 | 7.2 ± 1.5 | 4.621 | < **0.001** | 6.5 ± 1.6 | 6.7 ± 1.4 | 1.430 | 0.153 |

Abbreviations: ALB, serum albumin; ALT, alanine aminotransferase; AST, aspartate aminotransferase; BMI, body mass index; Cr, Creatinine; DB: direct bilirubin; DBP, diastolic blood pressure; FPG, fasting plasma glucose; Hb, hemoglobin; HDLC, high-density lipoprotein cholesterol; LDLC, low-density lipoprotein cholesterol; RBC, red blood cells; SBP, systolic blood pressure; TB: total bilirubin; TC, total cholesterol; TG, triglyceride; TP, serum total protein; UA, uric acid; UN, urea nitrogen; WBC, white blood cells.

three SNPs was significantly different between the male and female subjects (*P* > 0.05, S3 Table).

The genotype frequencies of the three tag SNPs of *SOD1* in normal and abnormal lipid subgroups of males were summarized in S4 Table. The distribution of the rs4998557 genotype were different between the low HDLC (< 1 mmol/L) subgroup and its normal control (≥1 mmol/L) subgroup (*P* = 0.03), as well as the rs17880487 genotype between the high TC (≥ 5.2 mmol/L) subgroup and its normal control (< 5.2 mmol/L) subgroup (*P* = 0.03). S5 Table showed the statistical analyses on the genotype frequencies of *SOD1* in the normal and abnormal lipid subgroups in females, and the distribution of the rs4998557 genotype in high TC individuals and the normal ones was statistically significant (*P* = 0.01). The lipid levels across the three tag SNPs in adult males was described in S6 Table, and the significant differences were found in TC and LDLC levels between CT and CC genotypes in additive model and between CT+TT and CC genotypes in dominant model of rs17880487 (*P* < 0.05). The LDLC levels between T and C alleles also presented a significant difference (*P* = 0.04). As shown in S7 Table, no differences were found for all the lipid parameters in any of the genotype models of the three tag SNPs in females (*P* > 0.05).

Further, logistic analyses were performed with adjustment for age, BMI, education, FPG, and smoking. Table 2 displayed the logistic regression analyses on the relationships between

**Table 2. Logistic regression analyses of lipids with three tag SNPs of superoxide dismutase 1 gene in adult males with adjustment for age, body mass index, education, fasting plasma glucose concentration, and smoking status[a, b].**

| Genotype | TG, mmol/L | | TC, mmol/L | | LDLC, mmol/L | | HDLC, mmol/L | |
|---|---|---|---|---|---|---|---|---|
| | $\geq$ 1.7 vs $<$ 1.7 | | $\geq$ 5.2 vs $<$ 5.2 | | $\geq$ 3.4 vs $<$ 3.4 | | $<$ 1.0 vs $\geq$ 1.0 | |
| Comparison | P | OR (95% CI) | P | OR (95% CI) | P | OR (95% CI) | P | OR (95% CI) |
| **rs4998557** | | | | | | | | |
| Add.: GG vs AG vs AA | 0.55 | 0.93 (0.73–1.18) | 0.17 | 0.85 (0.68–1.07) | **0.03** | 0.73 (0.54–0.98) | 0.08 | 1.91 (0.92–3.96) |
| Dom.: AA + AG vs GG | 0.64 | 0.91 (0.62–1.34) | 0.53 | 0.89 (0.63–1.27) | 0.30 | 0.79 (0.51–1.24) | 0.80 | 0.85 (0.25–2.87) |
| Rec.: AA vs AG + GG | 0.60 | 0.91 (0.64–1.29) | 0.15 | 0.79 (0.57–1.09) | **0.02** | 0.59 (0.38–0.93) | **0.02** | 2.78 (1.17–6.57) |
| Hom.: AA vs GG | 0.38 | 0.72 (0.35–1.50) | **0.03** | 0.46 (0.23–0.94) | 0.09 | 0.43 (0.16–1.13) | 0.76 | 0.59 (0.02–18.1) |
| Alle.: A vs G | 0.57 | 0.93 (0.73–1.19) | 0.27 | 0.88 (0.70–1.11) | **0.02** | 0.71 (0.52–0.96) | 0.18 | 1.62 (0.80–3.29) |
| **rs1041740** | | | | | | | | |
| Add.: CC vs CT vs TT | 0.79 | 1.04 (0.80–1.34) | 0.79 | 1.03 (0.82–1.31) | 0.06 | 1.34 (0.99–1.82) | 0.98 | 1.01 (0.50–2.02) |
| Dom.: CT + TT vs CC | 0.87 | 0.97 (0.69–1.36) | 0.41 | 0.88 (0.64–1.20) | 0.68 | 1.09 (0.73–1.62) | 0.35 | 1.67 (0.57–4.91) |
| Rec.: TT vs CT + CC | 0.43 | 1.21 (0.75–1.95) | 0.14 | 1.40 (0.90–2.18) | **0.01** | 2.19 (1.25–3.83) | 0.78 | 0.85 (0.28–2.63) |
| Hom.: TT vs CC | 0.43 | 1.37 (0.63–2.99) | 0.10 | 1.87 (0.88–3.97) | 0.073 | 2.52 (0.92–6.92) | 0.18 | 10.4 (0.33–324) |
| Alle.: T vs C | 0.76 | 1.04 (0.80–1.35) | 0.73 | 1.04 (0.82–1.33) | **0.048** | 1.37 (1.00–1.89) | 0.93 | 0.97 (0.48–1.96) |
| **rs17880487** | | | | | | | | |
| Add.: CC vs CT vs TT | 0.72 | 0.93 (0.61–1.41) | **0.047** | 0.67 (0.45–0.99) | 0.99 | 1.00 (0.62–1.63) | **0.02** | 3.08 (1.19–7.99) |
| Dom.: CT + TT vs CC | 0.80 | 0.94 (0.61–1.46) | **0.03** | 0.63 (0.42–0.95) | 0.88 | 0.96 (0.58–1.60) | 0.06 | 2.52 (0.97–6.56) |
| Rec: TT vs CT + CC | 0.61 | 0.53 (0.05–5.96) | 0.25 | 3.84 (0.39–38.1) | 0.14 | 4.66 (0.62–35.0) | 1.0 | 0.00 (0.00-NA) |
| Hom.: TT vs CC | 0.55 | 0.47 (0.04–5.48) | 0.31 | 3.33 (0.33–33.4) | 0.16 | 4.47 (0.57–35.3) | 1.0 | 0.00 (0.00-NA) |
| Alle.: T vs C | 0.81 | 0.95 (0.63–1.44) | 0.09 | 0.72 (0.49–1.06) | 0.78 | 1.07 (0.67–1.72) | 0.09 | 2.13 (0.89–5.10) |

[a] Abbreviations: Add., additive model; Alle., allelic model; Dom., dominant model; HDLC, high-density lipoprotein cholesterol; Hom., homozygous model; LDLC, low-density lipoprotein cholesterol; Rec., recessive model; SNPs, single nucleotide polymorphisms; TC, total cholesterol; TG, triglyceride.

[b] NA: not available due to 0 was found at least in one of the genotypes in either of the groups.

the lipids and the three tag SNPs of *SOD1* in adult males. Between the subgroups of TG (mmol/L) $\geq$ 1.7 and $<$ 1.7, no significances ($P > 0.05$) were found in all models of the three tag SNPs. Between the subgroups of TC (mmol/L) $\geq$ 5.2 and $<$ 5.2, the data presented significances ($P < 0.05$) in the homozygous model of rs4998557 ($P = 0.03$, OR = 0.46, 95% CI: 0.23–0.94) and the additive ($P = 0.047$, OR = 0.67, 95% CI: 0.45–0.99) and dominant ($P = 0.03$, OR = 0.63, 95% CI: 0.42–0.95) models of rs17880487, while other models of the genotype showed no significances ($P > 0.05$). Between the LDLC subgroups ($\geq$ 4.1 vs. $<$ 4.1 mmol/L), the additive ($P = 0.03$, OR = 0.73, 95% CI: 0.54–0.98), recessive ($P = 0.02$, OR = 0.59, 95% CI: 0.38–0.93), and allelic ($P = 0.02$, OR = 0.71, 95% CI: 0.52–0.96) models of rs4998557 and the recessive ($P = 0.01$, OR = 2.19, 95% CI: 1.25–3.83) and allelic ($P = 0.048$, OR = 1.37, 95% CI: 1.00–1.89) models of rs1041740 were significantly correlated with the risk of high LDLC. With regard to the HDLC subgroups ($\geq$ 1.0 vs. $<$ 1.0 mmol/L), the recessive model of rs4998557 ($P = 0.02$, OR = 2.78, 95% CI: 1.17–6.57) and the additive model of rs17880487 ($P = 0.02$, OR = 3.08, 95% CI: 1.19–7.99) revealed the significant contributions of the two SNPs to low HDLC. Table 3 displayed logistic regression analyses of lipids with the three tag SNPs of *SOD1* in adult females. No statistical differences were observed for any of the genotypic models in any of the lipid parameters.

## Discussion

It is well known that the mutations of some genes encoding proteins in the lipid metabolic process are fundamental causes of the rare familial dyslipidemia [2], while multiple genetic

**Table 3. Logistic regression analyses of lipids with three tag SNPs of superoxide dismutase 1 gene in adult females with adjustment for age, body mass index, education, fasting plasma glucose concentration, and smoking status[a, b].**

| Genotype | TG, mmol/L | | TC, mmol/L | | LDLC, mmol/L | | HDLC, mmol/L | |
|---|---|---|---|---|---|---|---|---|
| | ≥ 1.7 vs < 1.7 | | ≥ 5.2 vs < 5.2 | | ≥ 3.4 vs < 3.4 | | < 1.0 vs ≥ 1.0 | |
| Comparison | P | OR (95% CI) | P | OR (95% CI) | P | OR (95% CI) | P | OR (95% CI) |
| **rs4998557** | | | | | | | | |
| Add.: GG vs AG vs AA | 0.98 | 1.00 (0.74–1.35) | 0.52 | 0.93 (0.76–1.15) | 0.33 | 0.88 (0.67–1.14) | 0.15 | 0.60 (0.30–1.21) |
| Dom.: AA + AG vs GG | 0.95 | 1.02 (0.64–1.63) | 0.44 | 1.14 (0.82–1.58) | 0.36 | 0.83 (0.55–1.24) | 0.08 | 0.34 (0.10–1.12) |
| Rec.: AA vs AG + GG | 0.86 | 1.04 (0.67–1.63) | 0.10 | 0.77 (0.56–1.05) | 0.46 | 0.86 (0.58–1.28) | 0.56 | 0.74 (0.28–2.01) |
| Hom.: AA vs GG | 0.55 | 1.33 (0.52–3.39) | 0.40 | 0.75 (0.38–1.47) | 0.15 | 0.47 (0.16–1.32) | 0.26 | 0.12 (0.00–4.83) |
| Alle.: A vs G | 0.96 | 1.01 (0.74–1.38) | 0.53 | 0.93 (0.75–1.16) | 0.24 | 0.85 (0.65–1.12) | 0.09 | 0.51 (0.24–1.10) |
| **rs1041740** | | | | | | | | |
| Add.: CC vs CT vs TT | 0.50 | 1.12 (0.81–1.53) | 0.85 | 0.98 (0.79–1.22) | 0.43 | 1.12 (0.85–1.48) | 0.20 | 1.58 (0.78–3.21) |
| Dom.: CT + TT vs CC | 0.43 | 1.18 (0.78–1.80) | 0.46 | 0.90 (0.67–1.20) | 0.64 | 1.09 (0.76–1.58) | 0.054 | 3.18 (0.98–10.27) |
| Rec.: TT vs CT + CC | 0.88 | 1.04 (0.59–1.86) | 0.96 | 0.99 (0.66–1.49) | 0.43 | 1.23 (0.74–2.03) | 0.65 | 0.72 (0.17–2.97) |
| Hom.: TT vs CC | 0.82 | 0.90 (0.34–2.36) | 0.85 | 1.07 (0.52–2.18) | 0.21 | 1.98 (0.68–5.78) | 0.36 | 5.79 (0.14–239.42) |
| Alle.: T vs C | 0.49 | 1.12 (0.81–1.55) | 0.87 | 0.98 (0.78–1.23) | 0.41 | 1.13 (0.85–1.51) | 0.17 | 1.74 (0.79–3.83) |
| **rs17880487** | | | | | | | | |
| Add.: CC vs CT vs TT | 0.63 | 0.89 (0.55–1.43) | 0.78 | 0.95 (0.68–1.34) | 0.32 | 1.22 (0.83–1.80) | 0.32 | 1.58 (0.64–3.89) |
| Dom.: CT + TT vs CC | 0.71 | 0.91 (0.54–1.51) | 0.78 | 0.95 (0.66–1.36) | 0.54 | 1.14 (0.75–1.75) | 0.20 | 1.84 (0.72–4.71) |
| Rec: TT vs CT + CC | 0.60 | 0.54 (0.05–5.39) | 0.32 | 2.38 (0.43–13.12) | 0.07 | 4.38 (0.91–20.99) | 1.00 | 0.00 (0.00-NA) |
| Hom.: TT vs CC | 0.77 | 0.70 (0.07–7.39) | 0.37 | 2.24 (0.38–13.21) | 0.20 | 3.04 (0.55–16.79) | 1.00 | 0.00 (0.00-NA) |
| Alle.: T vs C | 0.51 | 0.85 (0.53–1.37) | 0.95 | 0.99 (0.71–1.38) | 0.24 | 1.26 (0.86–1.85) | 0.22 | 1.72 (0.72–4.09) |

[a] Abbreviations: Add., additive model; Alle., allelic model; Dom., dominant model; HDLC, high-density lipoprotein cholesterol; Hom., homozygous model; LDLC, low-density lipoprotein cholesterol; Rec., recessive model; SNPs, single nucleotide polymorphisms; TC, total cholesterol; TG, triglyceride.

[b] NA: not available due to 0 was found at least in one of the genotypes in either of the groups.

polymorphisms have contributions to the most nonfamilial dyslipidemia with complex molecular mechanisms. The previous animal studies demonstrated the antioxidant role of SOD1 to modulate the redox homeostasis in lipid metabolism [13–15], and cholesterol metabolism was affected by SOD1 even independent of its antioxidant activity via inhibiting the activity of 3-hydroxy-3-methylglutaryl CoA reductase and promoting the low-density lipoprotein receptor pathway in hepatocarcinoma cells [16]. SOD1 was also bound to almost all classes of circulating lipoproteins with relative high activity in low and high density lipoproteins [7], which partly underlay its potential in lipid metabolism. Having a possibility to affect the antioxidant or protein-interaction activities, the SNPs of *SOD1* may have associations with the lipid concentrations in circulation. Though there were reports on the associations of *SOD1* SNPs with obesity (rs2070424G [-251A/G]) [19], type 1 diabetes (rs2234694A [+35 A/C]) [20], type 2 diabetes (rs2234694C) [21], diabetic nephropathy (rs2234694C and rs1041740T) [22, 23], microalbuminuria (rs1041740T) [24], macroangiopathy (rs2234694C) [20], cardiovascular disease (rs36232792 [50-bp Ins/Del], rs1041740T, and rs17880487T) [25, 26], death from cardiovascular disease in patients with type 2 diabetes (rs9974610A, rs10432782G, and rs1041740T) [24], gastric cancer (rs4998557A) [36], Alzheimer's disease (rs2070424A) [32], hearing loss (rs4998557A) [37], cataract (rs2070424G) [38], peritonitis (rs1041740T) [39], and erysipelas (rs4998557G) [40], no human studies examined the relationship between the *SOD1* SNPs and the lipid profiles before our present study as far as we knew.

In our study, a significant difference in serum lipid profiles was observed between males and females. The males had a higher prevalence of dyslipidemia than the females, which was

consistent with the findings from the whole country [41, 42]. To explore the association of the *SOD1* SNPs with the lipid levels in males and females, each of the three tag SNPs (rs4998557, rs1041740, and rs17880487) were analyzed in additive, dominant, recessive, homozygous, and allelic models. In males, the homozygous model suggested rs4998557A to be a protective factor for high TC, and the additive, recessive, and allelic models suggested it to be a protective factor for high LDLC; but revealed by the recessive model, rs4998557A was a risk factor for low HDLC. For rs1041740, both the recessive and allelic models suggested that rs1041740T was a risk factor for high LDLC, while for rs17880487, the additive and dominant models demonstrated rs17880487T to be a protective factor for high TC. This implied some kind of consistency with the previous findings that rs1041740T was a risk factor [24, 26] and rs17880487T was a protective factor for cardiovascular diseases [26]. No association of the three tag SNPs of *SOD1* with TG was observed. Contrarily, in females, all the three tag SNPs showed no association with any of the lipid measures. Therefore, the association between the *SOD1* SNPs and the lipids had a phenomenon of sexual dimorphism. Interestingly, the correlation between SNPs and the sexual dimorphism of many diseases have been observed, and the gender differences in specific SNP-phenotype associations were extensively assessed [30].

The similar sexually dimorphic associations between the serum lipid levels with the SNPs of *ABCA1* rs2230808 [43], *ZNF259* rs2075290 [44], *SPTY2D1* rs7934205 [45], and *BCL7B* rs2237278 [46] were also reported previously. The mechanisms for the gender differences were much complex, including gene expressions and posttranslational effects in sexually dimorphic manners, sexual hormones determining phenotypic variations, and differentiation of external environment [47–49]. Due to factors such as physiological development and sex hormones, sex differences had impact on cardiometabolic diseases across life span [50]. Maybe some other parameters, significantly different between dyslipidemia and non-dyslipidemia in males but not in females (or *vice versa*), also contributed to the sexually dimorphic associations. For example, the platelet count was higher in the dyslipidemic males than in the non-dyslipidemic males, but no such phenomenon was observed in the two groups of females. The association of higher platelet count and risk of metabolic disorders was addressed in some studies [51–53], and it was supposed to be involved in the present sexual dimorphism in lipid metabolism to some extent.

The present study was conducted in the Chinese Han adults, so the genotypic findings may be different from those in the other ethnic groups. Though HWE test supported the representativeness of the sampled population from a health examination center for the three tag SNPs, participants were not selected by a strictly randomized sampling method, and the sample size was not very large, especially for the examination on the SNPs of low MAF. Finally, though age, BMI, etc. were used to adjust the logistic regression analysis, dietary intakes and physical activities are expected in the future work to adjust for more extensive factors potentially affecting the lipid metabolism.

## Conclusions

The gene polymorphisms of *SOD1* may affect the lipid (especially cholesterol) profiles of the adult Han Chinese males, but have no correlation with any of the lipid profiles in the females. This sexual dimorphism suggested a sex-specific consideration from the genetic aspect in the risk assessment of dyslipidemia.

## Supporting information

**S1 Fig. Selection of tag single nucleotide polymorphisms of superoxide dismutase 1 with Haploview software.** (A) Results of Tagger. (B) Linkage disequilibrium plot showing $r^2$

multiplying 100.
(DOCX)

**S2 Fig.** Genotyping the single nucleotide polymorphism of rs4998557 in superoxide dismutase 1 gene by melting curve analysis in molecular beacon probe-based qPCR experiment (A) and confirmation with Sanger sequencing (B). (A) Melting curve diagram generated by the original data collected with Roche 480 II qPCR instrument. (B) Sequencing analysis of the corresponding qPCR products with ▲ indicating the allelic loci.
(DOCX)

**S3 Fig.** Genotyping the single nucleotide polymorphism of rs1041740 in superoxide dismutase 1 gene by melting curve analysis in molecular beacon probe-based qPCR experiment (A) and confirmation with Sanger sequencing (B). (A) Melting curve diagram generated by the original data collected with Roche 480 II qPCR instrument. (B) Sequencing analysis of the corresponding qPCR products with ▲ indicating the allelic loci.
(DOCX)

**S4 Fig.** Genotyping the single nucleotide polymorphism of rs17880487 in superoxide dismutase 1 gene by melting curve analysis in molecular beacon probe-based qPCR experiment (A) and confirmation with Sanger sequencing (B). (A) Melting curve diagram generated by the original data collected with Roche 480 II qPCR instrument. (B) Sequencing analysis of the corresponding qPCR products with ▲ indicating the allelic loci.
(DOCX)

**S1 Table. Primers and molecular beacon probes to genotype the tag SNPs of human superoxide dismutase 1 gene.** [a] Abbreviations: 3'UTR, untranslated region; BP, backward primer; FP, forward primer; MAF, minor allele frequency; Pr, Probe; SNPs, single nucleotide polymorphisms. [b] The global MAF values were cited from 1000Genomes as indicated in the NCBI SNP database. [c] The bases in small letters is the adapter sequence to form the stem of the beacon probe. The base in bold underlined italics is one of the allele for the SNP. [d] "Yes" indicated the primer used for Sanger sequencing of the amplicon to confirm the genotypes judged by the melting curves in qPCR experiment.
(DOCX)

**S2 Table. Composition of dyslipidemia in male and female subjects.** [a] Abbreviations: HDLC, high-density lipoprotein cholesterol; LDLC, low-density lipoprotein cholesterol; TC, total cholesterol; TG, triglyceride. [b] Including those meeting the diagnosis criteria recommended by the Chinese guideline for dyslipidemia management* and those previously diagnosed as dyslipidemia patients. The prevalence of dyslipidemia between males and females was significantly different, $P < 0.001$. *Reference information: Joint committee for guideline revision. 2016 Chinese guidelines for the management of dyslipidemia in adults. J Geriatr Cardiol. 2018;15(1):1–29. Epub 2018/02/13. doi: 10.11909/j.issn.1671-5411.2018.01.011. PubMed PMID: 29434622; PubMed Central PMCID: PMCPMC5803534.
(DOCX)

**S3 Table. Comparisons of the genotype frequencies of three tag SNPs of superoxide dismutase 1 gene between adult males and females.** [a] Abbreviation: SNPs, single nucleotide polymorphisms.
(DOCX)

**S4 Table. Genotype frequencies of three tag SNPs of superoxide dismutase 1 gene in abnormal and normal lipid groups of adult males.** [a] Abbreviations: HDLC, high-density

lipoprotein cholesterol; LDLC, low-density lipoprotein cholesterol; SNPs, single nucleotide polymorphisms; TC, total cholesterol; TG, triglyceride. [b] Multiple comparisons revealed the significant difference of genotype frequencies of GA and AA between low and normal HDLC groups, $P < 0.05$. [c] Genotype frequencies of TT was too low and did not meet the criteria for Chi-square test, so genotype CT and TT were combined.
(DOCX)

**S5 Table. Genotype frequencies of three tag SNPs of superoxide dismutase 1 gene in abnormal and normal lipid groups of adult females.** [a] Abbreviations: HDLC, high-density lipoprotein cholesterol; LDLC, low-density lipoprotein cholesterol; SNPs, single nucleotide polymorphisms; TC, total cholesterol; TG, triglyceride. [b] Multiple comparisons revealed the significant difference of genotype frequencies of GA and AA between high and normal TC groups, $P < 0.05$. [c] Genotype frequencies of TT was too low and did not meet the criteria for Chi-square test, so genotype CT and TT were combined.
(DOCX)

**S6 Table. Comparison of lipid levels across genotype models of three tag SNPs of superoxide dismutase 1 gene in adult males.** [a] Abbreviations: Add., additive model; Alle., allelic model; Dom., dominant model; HDLC, high-density lipoprotein cholesterol; Hom., homozygous model; IQR, interquartile range; LDLC, low-density lipoprotein cholesterol; Rec., recessive model; SNPs, single nucleotide polymorphisms; TC, total cholesterol; TG, triglyceride. [b] Multiple comparisons revealed the significant difference between the genotypes of CT and CC, $P < 0.05$.
(DOCX)

**S7 Table. Comparison of lipid levels across genotype models of three tag SNPs of superoxide dismutase 1 gene in adult females.** [a] Abbreviations: Add., additive model; Alle., allelic model; Dom., dominant model; HDLC, high-density lipoprotein cholesterol; Hom., homozygous model; IQR, interquartile range; LDLC, low-density lipoprotein cholesterol; Rec., recessive model; SNPs, single nucleotide polymorphisms; TC, total cholesterol; TG, triglyceride.
(DOCX)

## Acknowledgments

We thank for the contributions of Xiaoying Zhou, Wenjing Nie, and Zheng Chen in performing the SNP measurement.

## Author Contributions

**Conceptualization:** Ji-Chang Zhou.

**Data curation:** Ping Xu, Yumei Zhu, Xiongshun Liang, Ji-Chang Zhou.

**Formal analysis:** Ping Xu.

**Funding acquisition:** Chunmei Gong, Ji-Chang Zhou.

**Investigation:** Yumei Zhu, Xiongshun Liang, Yuanfei Xu, Ji-Chang Zhou.

**Methodology:** Ping Xu, Yumei Zhu, Ji-Chang Zhou.

**Project administration:** Chunmei Gong, Changhua Huang, Ji-Chang Zhou.

**Resources:** Changhua Huang, Xiao-Li Liu.

**Supervision:** Yumei Zhu, Chunmei Gong, Changhua Huang, Xiao-Li Liu, Ji-Chang Zhou.

**Validation:** Ping Xu, Xiao-Li Liu.

**Writing – original draft:** Ping Xu, Ji-Chang Zhou.

**Writing – review & editing:** Ping Xu, Ji-Chang Zhou.

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
