## [Decision Letter · Decision Letter 0]

20 Mar 2020

PONE-D-19-26611

Genetic polymorphisms of superoxide dismutase 1 are associated with the serum lipid profiles of Han Chinese adults in a sexually dimorphic manner

PLOS ONE

Dear Dr. Zhou,

Thank you for submitting your manuscript to PLOS ONE. After careful consideration, we feel that it has merit but does not fully meet PLOS ONE’s publication criteria as it currently stands. Therefore, we invite you to submit a revised version of the manuscript that addresses the points raised during the review process.

The manuscript has been evaluated by two reviewers, and their comments are available below.

The reviewers have raised a number of major concerns. They feel the manuscript should outline a clearly-defined research question, and they request improvements to the reporting of methodological aspects of the study, for example, regarding the sequencing and statistical analyses. In addition, terms such as "mechanical correlations" and “certain inclusion criteria“ should be described in more detail.

Could you please carefully revise the manuscript to address all comments raised?

We would appreciate receiving your revised manuscript by May 04 2020 11:59PM. To enhance the reproducibility of your results, we recommend that if applicable you deposit your laboratory protocols in protocols.io, where a protocol can be assigned its own identifier (DOI) such that it can be cited independently in the future. For instructions see: http://journals.plos.org/plosone/s/submission-guidelines#loc-laboratory-protocols

We look forward to receiving your revised manuscript.

Kind regards,

Natasha Rickett

Academic Editor

PLOS ONE

Journal Requirements:

3. Please refer to any post-hoc corrections to correct for multiple comparisons during your statistical analyses. if these were not performed please justify the reasons. Also, please ensure you have thoroughly described your methodologies to the standard that they could be repeated. Phrases such as the blood being collected in "tubes with or without EDTA anticoagulation" create confusion.

Reviewers' comments:

Reviewer's Responses to Questions

**Comments to the Author**

1. Is the manuscript technically sound, and do the data support the conclusions?

Reviewer #1: Yes

Reviewer #2: No

2. Has the statistical analysis been performed appropriately and rigorously? 

Reviewer #1: Yes

Reviewer #2: No

3. Have the authors made all data underlying the findings in their manuscript fully available?

Reviewer #1: Yes

Reviewer #2: Yes

4. Is the manuscript presented in an intelligible fashion and written in standard English?

Reviewer #1: Yes

Reviewer #2: No

5. Review Comments to the Author

Reviewer #1: The articles by Xu et al. describe the SOD1 polymorphisms with the lipid disorders in the adult males and females of the Chinese.

This article is of interest but some points should be clarified:

Page 2, line 60: Is enough that the patients living in Shenzhen for > 2 years. This point is very important because of life style and it should be influenced in the results.

Page 2, line 61: How the authors demonstrated that the patients were free of liver diseases, renal diseases, and any cancers in the past 6 months. Please clarify this idea in this section.

Page 5, line 107: The authors must be described sequencing analyses.

Page 5, line 113: The authors must be added the OR value for the logistic regression analyses.

Page 5, section Data analysis: Some variable are showed in results, however it not describe in this section.

Page 5, line 118: The authors must be clarified the selection of males and female patients in this study. Please include this idea in the text.

Discussion section: Dyslipidemia is one of the most prevalent health problems in the modern era, and multiple factors are thought to be the etiology. In this study others etiology were evaluated?. The activity of 3-hydroxy-3-methylglutaryl CoA reductase was determined?. This point is very important due to these factors could be surrogate variable. Please clarify this idea

Reviewer #2: Abstract:

• Please explain the term "mechanical correlations".

• Please explain „certain inclusion criteria“.

Participants and sample collection:

• „The people visiting the health examination center were fasted overnight and did not take medicines to control the previously diagnosed dyslipidemia for more than 12 hours“.

The effects of most lipid-lowering agents, such as statins, persist for many days. I am afraid that 12 hours without lipid-lowering medication is definitely too short.

Results:

• The separation of the study participants into non-dyslipidemia and dyslipidemia groups does not make sense (Table 1). I would suggest to make a „lipid-lowering treatment group“ and a „no lipid-lowering treatment group“, and analyze associations of SOD1 SNPs with lipid parameters within these groups.

• Reducing continous parameters (e.g. cholesterol level) to binary parameters (Hypercholesterolemia yes/no) should not be done. This diminishes the statistical Power and complicates the evaluation of SOD1 allele effects (Table S4 and S5).

• Logistic regression analysis should be used for categorial outcome (e.g. dead/alive). I must not be used for metric variables such as cholesterol levels.

• As stated above, it makes no sense to analyse associations of SNPs with lipid levels without taking account of lipid-lowering treatment.

6. PLOS authors have the option to publish the peer review history of their article (what does this mean?). If published, this will include your full peer review and any attached files.

Reviewer #1: Yes: Yaxsier de Armas Rodríguez

Reviewer #2: No

---

## [Author Response · Author response to Decision Letter 0]

6 May 2020

Dear editor and reviewers,

Thank you very much for your critical and constructive comments!

Please find our responses in "Response to Reviewers" section.

Thanks!

---

## [Decision Letter · Decision Letter 1]

2 Jun 2020

Genetic polymorphisms of superoxide dismutase 1 are associated with the serum lipid profiles of Han Chinese adults in a sexually dimorphic manner

PONE-D-19-26611R1

Dear Dr. Zhou

We are pleased to inform you that your manuscript has been judged scientifically suitable for publication and will be formally accepted for publication once it complies with all outstanding technical requirements.

With kind regards,

Graham R. Wallace

Academic Editor

PLOS ONE

Additional Editor Comments (optional):

Reviewers' comments:

Reviewer's Responses to Questions

**Comments to the Author**

1. If the authors have adequately addressed your comments raised in a previous round of review and you feel that this manuscript is now acceptable for publication, you may indicate that here to bypass the “Comments to the Author” section, enter your conflict of interest statement in the “Confidential to Editor” section, and submit your "Accept" recommendation.

Reviewer #1: All comments have been addressed

Reviewer #2: All comments have been addressed

2. Is the manuscript technically sound, and do the data support the conclusions?

Reviewer #1: Yes

Reviewer #2: Yes

3. Has the statistical analysis been performed appropriately and rigorously? 

Reviewer #1: Yes

Reviewer #2: Yes

4. Have the authors made all data underlying the findings in their manuscript fully available?

Reviewer #1: Yes

Reviewer #2: Yes

5. Is the manuscript presented in an intelligible fashion and written in standard English?

Reviewer #1: Yes

Reviewer #2: Yes

6. Review Comments to the Author

Reviewer #1: (No Response)

Reviewer #2: (No Response)

7. PLOS authors have the option to publish the peer review history of their article (what does this mean?). If published, this will include your full peer review and any attached files.

Reviewer #1: Yes: Yaxsier de Armas Rodríguez

Reviewer #2: No

---

## [Editor Report · Acceptance letter]

10 Jun 2020

PONE-D-19-26611R1 

Genetic polymorphisms of superoxide dismutase 1 are associated with the serum lipid profiles of Han Chinese adults in a sexually dimorphic manner 

Dear Dr. Zhou:

I'm pleased to inform you that your manuscript has been deemed suitable for publication in PLOS ONE. Congratulations! Your manuscript is now with our production department. 

Kind regards, 

on behalf of

Dr. Graham R. Wallace 

Academic Editor

PLOS ONE